# A Comparative Analysis of the Piezoelectric Ultrasonic Appliance and Trephine Bur for Apical Location: An In Vitro Study

**DOI:** 10.3390/jpm11101034

**Published:** 2021-10-15

**Authors:** Esther Cáceres Madroño, Paulina Rodríguez Torres, Soraya Oussama, Álvaro Zubizarreta-Macho, María Bufalá Pérez, Jesús Mena-Álvarez, Elena Riad Deglow, Sofía Hernández Montero

**Affiliations:** 1Department of Implant Surgery, Faculty of Health Sciences, Alfonso X el Sabio University, 28691 Madrid, Spain; ecacemad@uax.es (E.C.M.); prodrtor@uax.es (P.R.T.); souss@myuax.com (S.O.); mperebuf@uax.es (M.B.P.); elenariaddeglow@gmail.com (E.R.D.); shernmon@uax.es (S.H.M.); 2Department of Surgery, Faculty of Medicine and Dentistry, University of Salamanca, 37008 Salamanca, Spain; 3Department of Endodontics, Faculty of Health Sciences, Alfonso X el Sabio University, 28691 Madrid, Spain; jmenaalv@uax.es

**Keywords:** apical location, apicoectomy, cone-beam computed tomography scan, piezoelectric, trephine bur

## Abstract

To compare and contrast the accuracy of piezoelectric ultrasonic insert (PUI) and trephine bur (TB) osteotomy site preparation techniques for apical location. (1) Material and methods: A total of 138 osteotomy site preparations were randomly distributed into one of two study groups. Group A: TB technique (*n* = 69) and B: PUI technique (*n* = 69). A preoperative cone-beam computed tomography scan and an intraoral scan were performed and uploaded to implant-planning software to plan the virtual osteotomy site preparations for apical location. Subsequently, the osteotomy site preparations were performed in the experimental models with both osteotomy site preparation techniques and a postoperative CBCT scan was performed and uploaded into the implant-planning software and matched with the virtually planned osteotomy site preparations to measure the deviation angle and horizontal deviation as captured at the coronal entry point and apical end-point between osteotomy site preparations using Student’s *t*-test statistical analysis. (2) Results: The paired *t*-test found statistically significant differences at the coronal entry-point deviations (*p* = 0.0104) and apical end-point deviations (*p* = 0.0104) between the TB and PUI study groups; however, no statistically significant differences were found in the angular deviations (*p* = 0.309) between the trephine bur and piezoelectric ultrasonic insert study groups. (3) Conclusions: The results showed that the TB is more accurate than the PUI for apical location.

## 1. Introduction

Periapical tissue damage as a result of the pulp necrosis process is considered the most common pathology found in the alveolar bone. The exposure of dental pulp to bacteria and their products (antigens) can lead to nonspecific inflammatory and specific immune responses in the surrounding root tissues, which can lead to periapical tissues damage [1,2,3,4]. Several treatment approaches have been proposed in an effort to prevent periapical diseases. Root canal treatment heals 86% of teeth, with 95% remaining asymptomatic and functional 4–6 years after root canal treatment [5]. Should root canal treatment fail, non-surgical endodontic retreatment is recommended; however, said retreatment may decrease success rates by up to 80% [6]. Endodontic microsurgery procedures are also recommended when non-surgical endodontic retreatment proves unsuccessful, or in cases in which it would prove impossible or have an unfavorable prognosis [7], such as when orthograde access to the apical area of the root canal system would be ineffective or technically impossible. In such cases, periapical surgery is a proposed option for removing the infectious process via curettage of the surrounding apical tissues, resectioning the apical portion of the tooth (apicoectomy) and preparing the root-end cavity for insertion of a retrograde filling material in order to preserve the tooth [8,9]. However, periapical surgery procedures often require excessive alveolar bone removal to enable apical location and periapical infection removal, which can impede the periapical tissue healing process [10]. As a result, novel radiographic techniques including cone-beam computed tomography (CBCT) scans can be used to diagnose and plan periapical surgical treatment, enabling predictable and conservative apical location, better periapical healing of the bone defect, shorter therapeutic time, and less postoperative discomfort without the risk of damaging adjacent anatomical structures [2,7,11]. Furthermore, the use of computer-assisted static navigation procedures using surgical templates guided by computer-aided design or computer-aided manufacturing (CAD/CAM) has improved the accuracy of osteotomy site preparation and prevalence of apical location compared to conventional free-hand techniques; however, these techniques require knowledge of CAD/CAM and entail longer treatment times and higher costs [7]. In addition, some osteotomy site preparation appliances are used to enable apical location, especially with non-fenestrated buccal cortical plates [1]. Osteotomy site preparations are traditionally performed using tungsten burs mounted on handpiece appliances and piezoelectric ultrasonic inserts (PUI) [2,12]. However, successful apical location and osteotomy site preparation are dependent on operator experience [12]. Advances in dental implant surgery are thus moving toward alternative treatment approaches for osteotomy site preparation in endodontic microsurgery using trephine bur (TB) appliances [7], which are widely used to remove failed implants and harvest bone grafts, as well as in apical location [13,14,15,16]. That being said, the accuracy of apical location using these surgical approaches has yet to be analyzed or assessed.

The present study aims to analyze, compare and contrast the accuracy of PUI and TB osteotomy site preparation techniques for apical location. The null hypothesis (H_0_) states that there is no difference in apical location between PUI and TB osteotomy site preparation techniques.

## 2. Materials and Methods

### 2.1. Study Design

Two hundred and twenty-four upper teeth from all dental sectors, which were extracted due to periodontal and orthodontic concerns, were selected for study at the Dental Centre of Innovation and Advanced Specialties at Alfonso X El Sabio University (Madrid, Spain), between February and April 2021. The sample size was selected according to a previous study [17] with a power effect of 88.4 (with anything above 80 being considered acceptable). In order to obtain statistically significant differences with a power effect of 80.00% to detect differences between the null hypothesis H_0_: μ_1_ = μ_2_ via a bilateral Student’s *t*-test of two independent samples, taking into account a significance level of 5.00%, 138 osteotomy site preparations were included in the study. The resulting manuscript for this study was written in accordance with the 2021 Preferred Reporting Items for Laboratory Studies in Endodontology (PRILE) guidelines [18,19]. In addition, the in vitro study was conducted as per the principles laid out in the German Ethics Committee’s statement on the use of organic tissues in medical research (Zentrale Ethikkommission, 2003). The study was authorized by the Ethics Committee of the Faculty of Health Sciences at University Alfonso X el Sabio (Madrid, Spain), in November 2020 (Process No. 28/2020). All patients gave their informed consent for their teeth to be transferred for use in the study.

### 2.2. Experimental Procedure

The teeth were embedded in 16 experimental models of epoxy resin (Ref. 20-8130-128, EpoxiCure^®^, Buehler, IL, USA) with 14 teeth each. One hundred and thirty-eight osteotomy site preparations were randomly distributed (Epidat 4.1, Galicia, Spain) into one of two study groups: group A. Trephine bur (Ref.: 330205486001, Antartica, Pleumeleuc, France) (TB) (*n* = 69) and B. Piezoelectric ultrasonic insert (Ref.: 05534700, W&H, Bürmoos, Austrich) (PUI) (*n* = 69). The teeth assigned to the experimental models presented the same anatomy and were positioned in the experimental model using a silicone splint to prevent different apical position between the different teeth of the experimental models. The silicone splint was created by conventional impression using a dental training model made of acrylic resin (Ref. 20-8130-128, EpoxiCure^®^, Buehler, IL, USA), which the teeth were subsequently placed on. The epoxy resin (Ref. 20-8130-128, EpoxiCure^®^, Buehler, IL, USA) was then mixed following the manufacture’s recommendations and poured into the silicone splint with the teeth. After the epoxy resin set, the silicone splint was removed from the epoxy resin model. In addition, the apical locations where osteotomy site preparations were performed were also randomly selected (Epidat 4.1, Galicia, Spain).

The experimental models of epoxy resin (Ref. 20-8130-128, EpoxiCure^®^, Buehler, IL, USA) were preoperatively scanned using cone-beam computed tomography (CBCT) (WhiteFox, Acteón Médico-Dental Ibérica S.A.U.-Satelec, Merignac, France) under the following exposure parameters: 105.0 kV peak, 8.0 mA, 7.20 s, and a 15 × 13-mm^2^ field of view. Next, a 3D surface scan was performed via 3D intraoral scan (True Definition, 3M ESPE™, Saint Paul, MN, USA) using 3D in-motion video imaging technology. Datasets obtained from the digital workflow were then uploaded to 3D implant planning software (NemoScan^®^, Nemotec, Madrid, Spain) prior to designing the virtual osteotomy site preparations for apical location. The virtual osteotomy site preparations randomly assigned to TB study group were designed with 3.5-mm diameter and 13.0-mm length by matching the data from the 3D surface scan and CBCT, aligning the key points identified on the crowns of the teeth. Virtual osteotomy site preparations were designed until the apex of each tooth and at an insertion angle of 90° in relation to the teeth’s longitudinal axes. Virtual osteotomy site preparations randomly assigned to PUI study group were designed with 3.3-mm diameter and 13.0-mm length by matching the data from the 3D surface scan and CBCT, aligning the key points identified on the crowns of the teeth. Virtual osteotomy site preparations were designed until the apex of each tooth and at an insertion angle of 90° in relation to the teeth’s longitudinal axes (Figure 1).

Osteotomy site preparations for apical location of both study groups were performed by a different operator for each group, following the manufacturer’s recommendations.

### 2.3. Measurement Procedure

After carrying out osteotomy site preparations for apical location, postoperative CBCT scans of the experimental models were taken (Figure 2A–D).

Planned virtual osteotomy site preparations and postoperative CBCT scans were taken of both study groups and subsequently uploaded to an 3D implant planning software (NemoScan^®^, Nemotec, Madrid, Spain). These data were then matched in order to record deviation angle (taken in the center of the cylinder) and horizontal deviation (taken at the coronal entry-point and apical end-point). The measurements were taken by an independent observer (Figure 3A,B).

### 2.4. Statistical Tests

All variables of interest were input into SPSS 22.00 for Windows for statistical analysis with SPSS. Descriptive statistical analysis of quantitative variables was expressed as the mean and standard deviation (SD) of said variables. Comparative analysis was carried out by assessing difference in mean deviation between planned and performed osteotomy site preparations for the apical location between TB and PUI study groups using Student’s *t*-test (as the variables were normally distributed); *p* < 0.05 was defined as statistically significant.

## 3. Results

Table 1 shows the means and SD values of the coronal entry point (mm), apical end point (mm) and angular deviation (°) of planned and performed osteotomy preparations for the apical location between TB and PUI study groups.

The paired *t*-test found statistically significant differences in the coronal entry point deviations of the planned and performed osteotomy site preparations for the apical location between TB and PUI study groups (*p* = 0.0104) (Figure 4).

Moreover, the paired *t*-test also found statistically significant differences in the apical end-point deviations of the planned and performed osteotomy site preparations for the apical location between TB and PUI study groups (*p* = 0.004) (Figure 5).

Finally, the paired *t*-test found no statistically significant differences in the angular deviations of planned and performed osteotomy site preparations for the apical location between TB and PUI study groups (*p* = 0.309) (Figure 6).

## 4. Discussion

The results of the present study reject the null hypothesis (H_0_) that states that there is no difference in apical location between TB and PUI osteotomy site preparation techniques.

The TB technique enabled more accurate osteotomy preparations at the coronal entry-point and apical end-point than the PUI technique for the apical location. In addition, the osteotomy preparations performed using the TB technique were more regular than the PUI technique.

Many authors have evaluated the accuracy of various therapeutic procedures with this methodological procedure, such as the accuracy of dental implant placement using computer-aided static navigation techniques, which showed a 0.99-mm horizontal deviation (ranging between 0.0–6.5 mm) at the dental implant platform, 1.24-mm horizontal deviation (ranging between 0.0–6.9 mm) at the dental implant apex, and an average angle deviation of 3.81° (ranging between 0.0–24.0°) relative to the longitudinal axis of dental implants [20,21]. The accuracy range provided to dental implants has led to computer-aided static navigation techniques being applied in other dental disciplines such as root canal location, which showed a statistically significant difference between computer-aided static navigation techniques and manual access cavities at the coronal (*p* < 0.0001), apical (*p* < 0.0001), and angular (*p* < 0.0001) levels [17]. Both root canal and apical location require high accuracy rates due to the reduced working field and high risk of intraoperative complications; therefore, several studies have been conducted in an effort to determine the most accurate computer-aided navigation technique. The apical location of the root apex through conservative surgical access cavities positively impacts the outcome of periapical healing of any bone defects, operating time, accuracy, and level of postoperative discomfort, without the risk of damaging the surrounding structures [2]. Therefore, clinicians should consider using drilling guided by computer-aided static navigation techniques, especially when there is compromised surgical access, with these techniques resulting in limited periapical tissue damage and no cortical loss despite limited vision of resected roots and difficulty inserting and orienting ultrasonic tips along the longitudinal axis of the tooth [1]. However, the inaccuracy of computer-aided navigation techniques, in addition to high costs, longer times and steep learning curves, may contribute to clinicians continuing to use conventional free-hand techniques for surgical procedures without the assistance of computer-aided navigation techniques.

Root apex location presents a major challenge for clinicians performing endodontic microsurgical procedures [22]. Magnification, illumination, microinstruments, and CBCT scans are traditionally used in endodontic microsurgical procedures to improve the success rate of root apex location, but computer-assisted static navigation techniques have shown a 27 times higher success rate in root apex location than conventional endodontic microsurgical procedures [23]. Furthermore, the success rate of apical location was established at 96.8% (confidence interval of 93.0% to 100%) of the procedures carried out using a computer-assisted static navigation technique; therefore, computer-assisted static navigation techniques are highly recommended to help locate the apical root in endodontic microsurgical procedures. In addition, computer-assisted static navigation techniques for root apex location are usually planned with trephine burs, because the cylindrical geometry of the trephine bur prevents undesirable deviations during drilling. On the other hand, if the root apex is not located using computer-aided navigation techniques, or if osteotomy preparation does not favor root apex resection and/or root-end cavity preparation, osteotomy preparation must be continued using conventional free-hand techniques with PUI because the latter provides clinicians the ability to relocalize the direction of the osteotomy preparation more conservatively than the TB technique. Moreover, conventional free-hand techniques are especially recommended in cases of patients with limited mouth opening or posterior region treatments in which the surgical splint is difficult to insert [24,25].

However, no technology has been found capable of guaranteeing 100% success in locating the root apex, and thus new technologies have been tested for the accurate location of the root apex. Gambarini et al. reported a clinical case in which computer-assisted dynamic navigation techniques were used for apical location in endodontic microsurgery. These techniques employ an optical triangulation tracking system with stereoscopic motion-tracking cameras that guide the drilling process in real time, ensuring that the planned angle, trajectory and depth of the osteotomy are achieved [22]. These techniques are widely used in dental implant placement, with significantly lower deviation values (*p* ˂ 0.05) at the coronal entry point (0.71 ± 0.40 mm), apical end point (1.00 ± 0.49 mm) and angular deviation (2.26 ± 1.62°) when compared with traditional freehand dental implant placement techniques [26,27]. Additionally, computer-aided dynamic navigation techniques are now used in the field of endodontics to improve the accuracy of root canal location and circumvent the potential risks of these treatments. [11,28,29]. However, a recent systematic review and meta-analysis found no statistically significant differences between the root canal location success rates of static and dynamic computer-aided navigation techniques (*p* = 0.185) [30]. Moreover, statistically significant differences were found between computer-aided static navigation techniques and conventional free-hand techniques for apical location (*p* < 0.0001) [23]. Nonetheless, conventional freehand techniques are still widely used, and several articles report success rates of apical location using piezoelectric ultrasonic insert and trephine bur appliances [13,14,15,16]. Further studies are, therefore, necessary to provide information about the accuracy of the conventional freehand techniques for apical location.

This in vitro study is potentially limited due to its experimental nature; for example, similarities in tooth anatomy and the dental position of teeth may differ from a real clinical situation. However, the teeth were selected based on anatomy and were also randomized. Furthermore, the silicone splint enabled a repeatable dental position across all experimental models. In addition, the methodological procedure used in this study is easily applicable to clinical studies.

## 5. Conclusions

Bearing in mind the limitations of this in vitro study, the results of the present study found that the trephine bur is more accurate than piezoelectric ultrasonic insert for apical location.

## Figures and Tables

**Figure 1 jpm-11-01034-f001:**
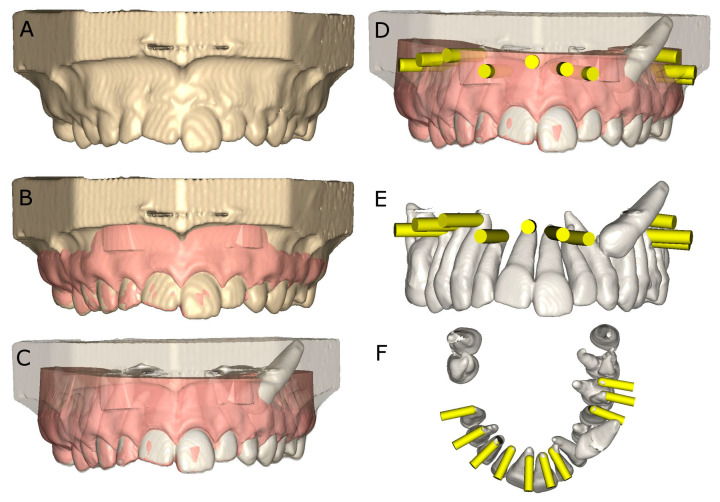
(**A**) CBCT scan rendering, (**B**) alignment procedure between STL and CBCT scan digital files, (**C**) transparent rendering of the CBCT scan, (**D**) front view of the virtual osteotomy site preparation planning (yellow cylinders), with surrounding tissues, (**E**) front view and (**F**) apical view of the virtual osteotomy site preparation planning (yellow cylinders), without surrounding tissues.

**Figure 2 jpm-11-01034-f002:**
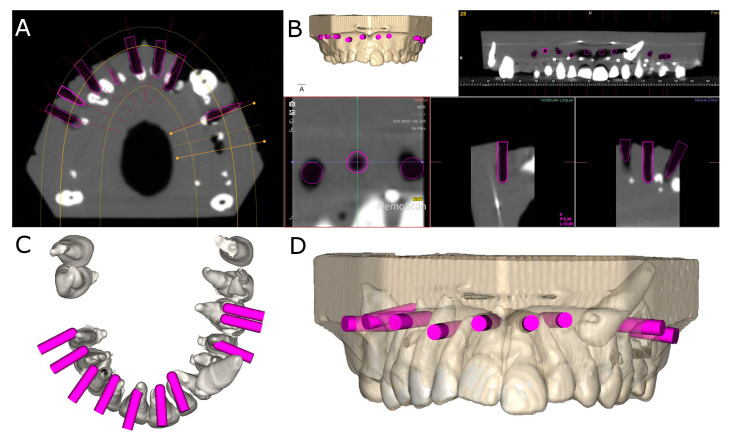
(**A**) Postoperative CBCT scan, (**B**) CBCT scan rendering, (**C**) apical view and (**D**) front view of the osteotomy site preparations (pink cylinders).

**Figure 3 jpm-11-01034-f003:**
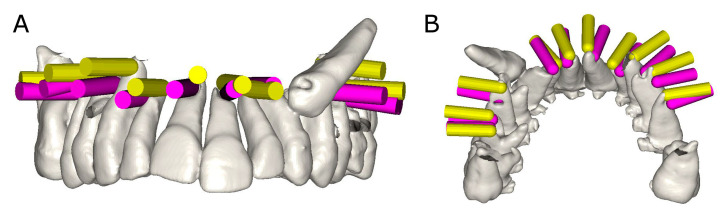
(**A**) Front view and (**B**) apical view of the deviation measurement procedure between planned (yellow cylinder) and performed (pink cylinder) osteotomy site preparations.

**Figure 4 jpm-11-01034-f004:**
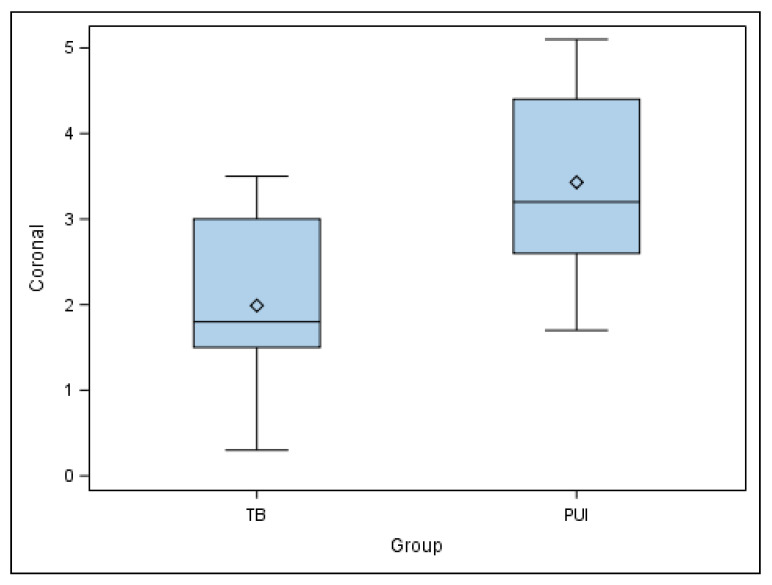
Box plot of coronal deviations of planned and performed osteotomy site preparations for apical location between TB and PUI study groups.

**Figure 5 jpm-11-01034-f005:**
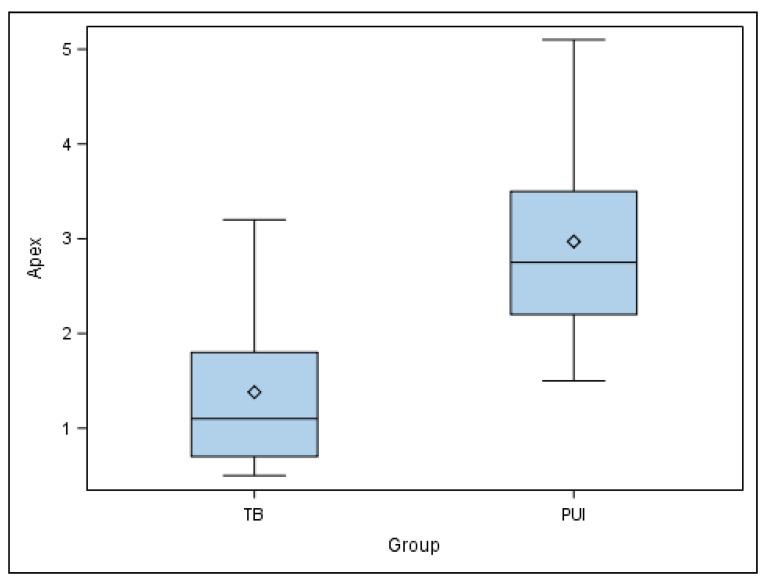
Box plot of apical deviations of planned and performed osteotomy site preparations for the apical location between TB and PUI study groups.

**Figure 6 jpm-11-01034-f006:**
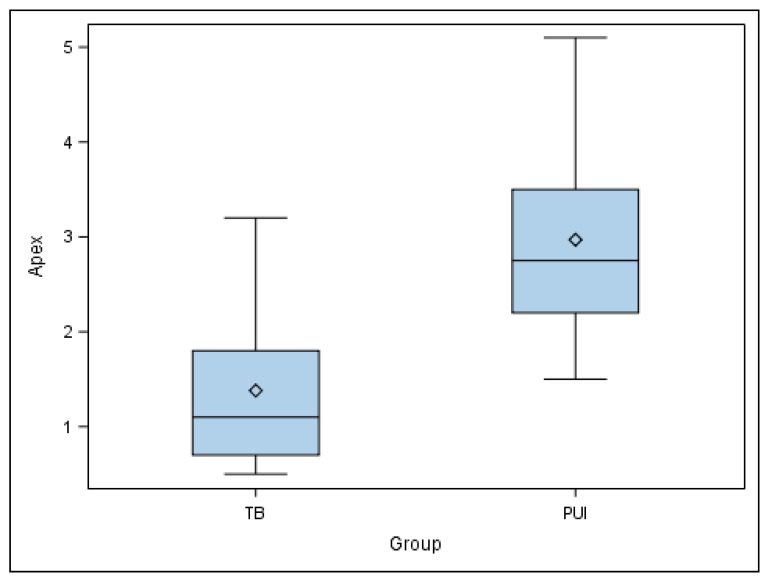
Box plot of angular deviations of planned and performed osteotomy site preparations for the apical location between TB and PUI study groups.

**Table 1 jpm-11-01034-t001:** Descriptive deviation values at the coronal entry point (mm), apical end point (mm), and angular (°) levels of planned and performed osteotomy preparations for apical location between TB and PUI study groups.

		*n*	Mean	SD	Minimum	Maximum
Coronal	TB	69	1.99 ^a^	1.03	0.30	3.50
	PUI	69	3.43 ^b^	1.21	1.70	5.10
Apical	TB	69	1.38 ^a^	0.87	0.50	3.20
	PUI	69	2.97 ^b^	1.26	1.50	5.10
Angular	TB	69	10.96 ^a^	4.23	2.30	16.60
	PUI	69	9.02 ^a^	4.07	2.80	15.50

^a,b^ Statistically significant differences (*p* < 0.05) between groups.

## Data Availability

Select data can be granted on request as per any relevant restrictions (e.g., privacy or ethical).

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
