# Peer review of "A Comparative Analysis of the Piezoelectric Ultrasonic Appliance and Trephine Bur for Apical Location: An In Vitro Study"

_jpm, 2021, doi:10.3390/jpm11101034_

Round 1
Reviewer 1 Report
Dear Authors, thank you for submitting your paper.
The aim of the present study is to analyze, compare and contrast the accuracy of PUI and TB osteotomy site preparation techniques for apical location. The null hypothesis (H0) states that there is no difference in apical location between PUI and TB osteotomy site preparation techniques
I congratulate the authors for this very relevant research.
It appears well structured, correctly carried out and written without logical or factual errors
Methodological aspects are deeply cleared in the manuscript.
The topic is in line with the journal aim.
-Data reported in the Methods section are appropriate and precisely described;
Typos are present in the manuscript, please correct them
-Results are reported clearly and adequately supported by Tables.
-I suggest to the Authors to improve their reference list citing the following recent article: https://doi.org/10.3390/children8010033
The Conclusions are correctly stated and supported by the findings obtained from the present study.
According to this Reviewer’s consideration, novelty and quality of the paper, publication of the present manuscript is recommended.
Author Response
Dear Reviewer 1:
I’m pleased to resubmit the manuscript of the work entitled, "Comparative Analysis of the Piezoelectric Ultrasonic Appliance and Trephine Bur for Apical Location. An In Vitro Study”
Reviewer 1: Typos are present in the manuscript, please correct them
Response: In order to adapt to the reviewer's 1 comments, we have corrected the typo errors.
Reviewer 1: -I suggest to the Authors to improve their reference list citing the following recent article: https://doi.org/10.3390/children8010033
Response: In order to adapt to the reviewer's 1 comments, we have cited the mentioned reference.
We take this opportunity to thank the recommendations and suggestions made by the reviewers to improve the document.
Yours sincerely,
Reviewer 2 Report
In my opinion, the present manuscript provides clear and pertinent data, within the limitations of an in vitrostudy, with good quality of presentation. I, therefore, recommend its acceptance, after the following minor revisions:
- There is a discrepancy between the lines 80-81 (”The null hypothesis (H0) states thatthere is no differencein apical location between PUI and TB osteotomy site preparation techniques.”) and lines 189-190 (”The results of the present study reject the null hypothesis (H0) that states that there is differencein apical location between TB and PUI osteotomy site preparation techniques.”).
- Lines 253-255 require references.
Author Response
Dear Reviewer 1:
I’m pleased to resubmit the manuscript of the work entitled, “Comparative Analysis of the Piezoelectric Ultrasonic Appliance and Trephine Bur for Apical Location. An In Vitro Study”.
Reviewer 2: There is a discrepancy between the lines 80-81 (”The null hypothesis (H0) states thatthere is no differencein apical location between PUI and TB osteotomy site preparation techniques.”) and lines 189-190 (”The results of the present study reject the null hypothesis (H0) that states that there is differencein apical location between TB and PUI osteotomy site preparation techniques.”).
Response: In order to adapt to the reviewer's 2 comments, we have changed the null hypothesis sentence of the Discussion section.
Reviewer 2: Lines 253-255 require references.
Response: In order to adapt to the reviewer's 2 comments, we have added references.
We take this opportunity to thank the recommendations and suggestions made by the reviewers to improve the document.
Yours sincerely,